# Structural distortion and electron redistribution in dual-emitting gold nanoclusters

Qi Li[1,2,6], Dongming Zhou[3,6], Jinsong Chai[1], Woong Young So [1], Tong Cai [4], Mingxing Li[5], Linda A. Peteanu[1], Ou Chen [4], Mircea Cotlet[5], X. Wendy Gu [2✉], Haiming Zhu [3✉] & Rongchao Jin [1✉]

Deciphering the complicated excited-state process is critical for the development of luminescent materials with controllable emissions in different applications. Here we report the emergence of a photo-induced structural distortion accompanied by an electron redistribution in a series of gold nanoclusters. Such unexpected slow process of excited-state transformation results in near-infrared dual emission with extended photoluminescent lifetime. We demonstrate that this dual emission exhibits highly sensitive and ratiometric response to solvent polarity, viscosity, temperature and pressure. Thus, a versatile luminescent nanosensor for multiple environmental parameters is developed based on this strategy. Furthermore, we fully unravel the atomic-scale structural origin of this unexpected excited-state transformation, and demonstrate control over the transition dynamics by tailoring the bi-tetrahedral core structures of gold nanoclusters. Overall, this work provides a substantial advance in the excited-state physical chemistry of luminescent nanoclusters and a general strategy for the rational design of next-generation nano-probes, sensors and switches.

[1] Department of Chemistry, Carnegie Mellon University, Pittsburgh, PA 15213, USA. [2] Department of Mechanical Engineering, Stanford University, Stanford, CA 94305, USA. [3] Centre for Chemistry of High-Performance and Novel Materials, Department of Chemistry, Zhejiang University, 310027 Hangzhou, Zhejiang, China. [4] Department of Chemistry, Brown University, Providence, RI 02912, USA. [5] Center for Functional Nanomaterials, Brookhaven National Laboratory, Upton, NY 11973, USA. [6] These authors contributed equally: Qi Li, Dongming Zhou. ✉email: xwgu@stanford.edu; hmzhu@zju.edu.cn; rongchao@andrew.cmu.edu

Photoexcitation can alter molecular structure and electron distribution[1–17]. This interesting phenomenon has endowed some organic molecules and metal complexes with two drastically different states, which are particularly appealing for applications in sensing, probes, switches, and actuators[1–17]. However, this combined excited-state process has not yet been identified in nanomaterials, which is probably because of the increased structural rigidity and enhanced electron delocalization as materials transition from the molecular scale to the nanoscale. In the past decade, ultrasmall nanoparticles of 1–3 nm in diameter (often called nanoclusters) have attracted attention in the nanoscience community due to the atomic precision achieved in their synthesis and characterization and fascinating optical properties, such as the near-infrared (NIR) photoluminescence (PL)[18–29]. We propose that nanoclusters can combine the merits of both larger nanoparticles (e.g., the longer fluorescence lifetime) and smaller molecules (e.g., the rich excited-state transformations), which can provide an exclusive opportunity for the development of new functional materials with synergetic properties.

Herein we directly identify a significant structural distortion accompanied by an electron redistribution in three photoexcited atomically precise nanoclusters ($Au_{24}(S\text{-}TBBM)_{20}$[30], $Au_{14}Cd_1(S\text{-}Adm)_{12}$[31], and $Au_{24}(S\text{-}PET)_{20}$[32], where S-TBBM = 4-tertbutyl-phenylmethancan, S-Adm = 1-adamantanethiol, and S-PET = 2-phenylethanethiol) using a combination of different spectroscopic techniques. The photo-induced changes in the structure and electron distribution in these nanoclusters lead to two significantly different excited states, thus dual emission is observed. The dual emission in these Au nanoclusters shows significant sensitivity to solvent polarity, viscosity, temperature, and pressure. Such dual emission with high sensitivity to multiple environmental parameters, together with the NIR II emission and the long PL lifetimes, are features that indicate that these Au nanoclusters are promising as next-generation probes and sensors. Furthermore, as the atomic structures of Au nanoclusters have been totally solved by the single-crystal X-ray diffraction (XRD)[18,30,31], the atomic-scale structural origin of such excited-state transformation (structural distortion with electron redistribution) can now be fully unraveled. We find that the bi-tetrahedral core in Au nanoclusters is the fundamental structural origin of the excited-state structural distortion with electron redistribution, and the flexibility of the individual tetrahedra govern the excited-state transformation dynamics.

## Results

**Dual emission and excited-state dynamics of $Au_{24}$.** Figure 1 displays the atomic structure, steady-state optical spectra and PL dynamics of $Au_{24}(S\text{-}TBBM)_{20}$ ($Au_{24}$ for short hereafter.) Optical data were measured in dichloromethane (DCM). The atomic structure of $Au_{24}$ (Fig. 1a) possesses a bi-tetrahedral $Au_8$ core, in which the two tetrahedra are anti-prismatically joined together through two triangular faces (face to face). This bi-tetrahedral $Au_8$ core is protected by four $Au_4S_5$ surface motifs. The ultraviolet–visible (UV-Vis) absorption (black) and PL (gray) spectrum of $Au_{24}$ are shown in Fig. 1b. In the absorption spectrum, a peak at 500 nm with a shoulder at 420 nm can be observed. Based on previous density functional theory (DFT) calculations, the 500 nm peak can mainly be attributed to the highest occupied molecular orbital–lowest unoccupied molecular orbital (HOMO–LUMO) transition, with a small contribution from the HOMO-2 to LUMO transition[30]. Both the HOMO and LUMO orbitals are distributed around the bi-tetrahedral $Au_8$ core; thus the HOMO–LUMO transition occurs within the $Au_8$ core[30]. The PL spectrum of $Au_{24}$ (Fig. 1b, gray) in DCM shows

dual emission, with one visible PL at 670 nm and one NIR PL at 1050 nm. The overall quantum yield (QY) of $Au_{24}$ is ~2% using an integrating sphere. The PL excitation spectra for the two emission bands (Fig. 1c) were also measured. Both PL excitation spectra are similar as the UV-Vis absorption spectrum with the major peak at ~500 nm, indicating that both PL bands are excited by the $Au_8$-core-based HOMO–LUMO transition.

The PL dynamics was studied by the time-correlated single-photon counting (TCSPC) technique (Fig. 1d–f). Two lifetime components are required to fit the decays of PL I (Fig. 1d, detected from 550 to 750 nm) and PL II (Fig. 1e, detected from 900 to 1000 nm). The components of 1.6 (82%) and 77.1 ns (18%) are required for PL I, and longer components of 66 ns (77%) and 263 ns (23%) are required for PL II. Interestingly, when zooming into the early 2-ns range (Fig. 1f), it can be observed that the fast decay of PL I corresponds to the rise of the PL II. This correspondence strongly suggests the direct electron transfer from PL I state to PL II state. This is the first time, to the best of our knowledge, that such a direct transfer event has been reported in metal nanoclusters.

To further unravel the excited-state dynamics of $Au_{24}$ in DCM, a femtosecond transient absorption (fs-TA) study was conducted (Fig. 2 and Supplementary Fig. 1). The fs-TA spectra of $Au_{24}$ at typical time delays within the first 1.4 ns are displayed in Supplementary Fig. 1. Rich electronic dynamics were observed within the 1.4-ns time window (the limit of our set-up). As shown in Fig. 2a, b, after pumping $Au_{24}$ at 520 nm, a ground-state bleaching (GSB) signal at ~500 nm was immediately observed, which corresponds to the HOMO–LUMO transition. This GSB signal remains constant during the entire time window, indicating that no excited electrons relax back to the ground state in the 1.4-ns period. On the other hand, three excited-state absorptions (ESAs) can be identified between 550 and 1300 nm (the red end of our window) and their time-dependent evolutions are significantly different. Starting from ~100 fs to 1 ps, one can observe a decay of the ESA centered at ~600 nm and the rise of another ESA from 750 to 1200 nm (kinetic traces are displayed in Fig. 2b). More interestingly, starting from ~10 ps, one can observe another significant electron transition process, which lasts until the end of our time-delay window (1.4 ns). It can be observed that there is a decrease of ESA from 750 to 1200 nm, accompanied by the rise of a new ESA from 600 to 850 nm. This nanosecond process during the TA measurement is consistent with the early time kinetics (within 1.5 ns) observed in TCSPC measurements (Fig. 1f). The relaxation diagram of $Au_{24}$ is displayed in Fig. 2c.

**Ratiometric PL response to multiple environmental parameters**. To obtain a deeper insight of the underlying mechanism of the dual emission and the state-to-state transition, the PL properties of $Au_{24}$ were systematically studied in different environments (polarity, viscosity, solid state, and temperature). First, we describe the properties of $Au_{24}$ in solvents of different polarity. The PL properties of $Au_{24}$ in DCM, toluene, and hexane are shown in Fig. 3a. The overall QY of $Au_{24}$ increases from 2% in DCM to 10% in hexane. It can be observed that, from DCM to hexane (polarity decreases), the PL I significantly increases while the PL II slightly decreases. The UV-Vis absorption and PL excitation spectra of $Au_{24}$ in hexane are similar to those in DCM, indicating no change of the ground state (Supplementary Fig. 2a, b). The lifetime of PL I shows a drastic increase to >200 ns from DCM to hexane, and the fast sub-nanosecond decay that is observed in DCM disappears in hexane (Fig. 3b). In DCM, the lifetime of the PL I state is predominantly determined by this sub-nanosecond non-radiative decay, which arises from the state-to-state transition. Transferring $Au_{24}$ from DCM to low-polarity

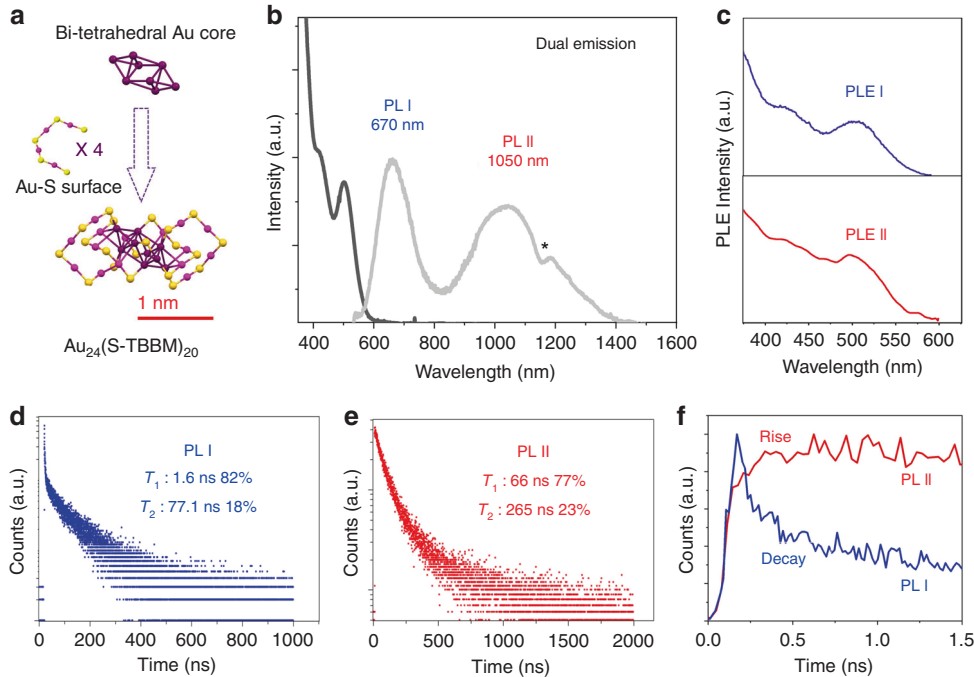

**Fig. 1 Optical properties and PL dynamics of Au$_{24}$. a** Anatomy of the atomic structure of Au$_{24}$ determined by single-crystal X-ray diffraction[30]. Purple = core Au; magenta = surface Au; yellow = S; Carbon tails are omitted for clarity. **b** UV-Vis absorption (black) and PL (gray) spectra (the asterisk denotes the spectrum "structuration", which is induced by the solvent). **c** PL excitation spectra for the two emissions, measured at 650 nm (blue) and 1050 nm (red), respectively. **d, e** Time-correlated single-photon counting (TCSPC) trajectories of the PL I (detected from 550 to 750 nm) and PL II (detected from 900 to 1000 nm). **f** Comparison of the TCSPC trajectories of PL I (blue) and PL II (red) in the early 1.5 ns. PL photoluminescence, $\tau$ PL lifetime component.

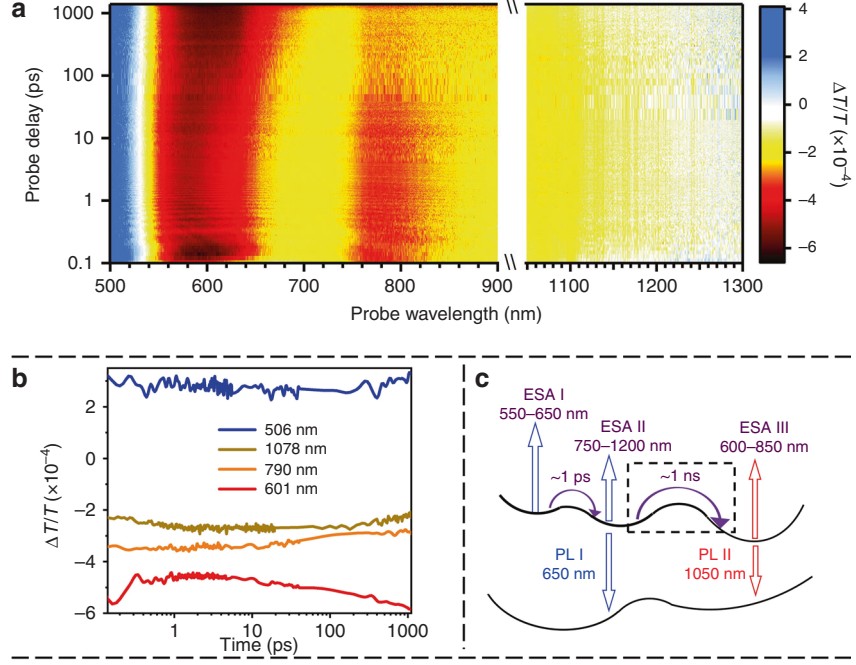

**Fig. 2 Excited-state dynamics of Au$_{24}$. a** Map of the TA spectra of Au$_{24}$ pumped at 520 nm, which shows the $\Delta T/T$ at all time delays between 500 and 900 nm and between 1050 and 1300 nm. **b** Selected kinetic traces at typical wavelengths. **c** Relaxation diagram for Au$_{24}$. Arrows denote the transitions between different electronic states. PL photoluminescence, ESA excited state absorption.

solvents greatly slows down this process and thus significantly increases the overall PL efficiency of Au$_{24}$.

We have also measured the PL spectra of Au$_{24}$ in other solvents and in the solid state, as well as under liquid nitrogen. As shown in Fig. 3c and Supplementary Figs. 3–5, significantly different PL spectra can be observed in these environments. From DCM to

butanol, the QY of Au$_{24}$ increases from 2% to 20%, and no nanosecond decay can be observed in the TCSPC spectrum (Supplementary Fig. 3). QY up to 25% can be obtained in the solid state (crystal or film, Fig. 3c) and further increases to ~80% in liquid nitrogen (Supplementary Fig. 4). From DCM to liquid nitrogen, the significant increase (~100 times) and blue shift

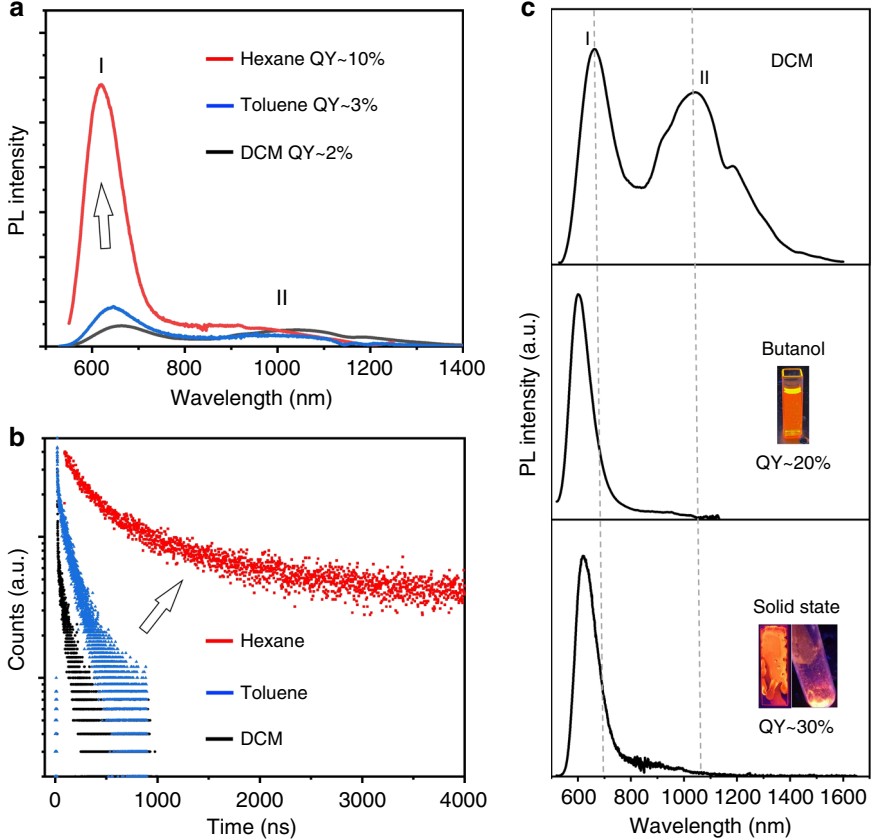

**Fig. 3 Solvent polarity and viscosity dependence, as well as the solid-state PL of Au$_{24}$.** Comparison of the PL (**a**) and TCSPC (**b**) of Au$_{24}$ in DCM (black), toluene (blue) and hexane (red). TCSPC trajectories were detected from 550 to 650 nm. **c** Comparison of the PL of Au$_{24}$ in DCM (top), butanol (middle), and solid state (crystal or film, bottom). From top to bottom, the overall QY of Au$_{24}$ increases from 2% to 30%, which is accompanied by an increase in the ratio of PL I/PL II and the blue shift of the PL I. Inset photographs: Au$_{24}$ in butanol and solid-state film and single crystals under UV lamp. QY quantum yield.

(from 650 to 600 nm) of PL I are observed, while the PL II almost remains unchanged and thus becomes negligible relative to PL I. This result suggests that the nanosecond transition from PL I state to PL II state is sensitive to the external environment.

Supplementary Table 1 summarizes the position of PL I in different environments. The first major conclusion that can be drawn from this data is that the PL of Au$_{24}$ is dependent on the polarity of the solvent. The more polar the solvent, the more red-shifted the PL (both I and II), and the stronger the PL II intensity. This polarity dependence suggests the charge-transfer nature in both PL states. The second conclusion is that the PL of Au$_{24}$ is dependent on the solvent viscosity. In solvents with higher viscosity (i.e., butanol, octanol), significant enhancement of QY and increase of the PL I/PL II ratio were observed. And by adding more viscous solvent, the 1,2-dichlorobenzene into the DCM solution of Au$_{24}$, it can be clearly observed that the nanosecond state transition gradually slows down (Supplementary Fig. 5). Such viscosity dependence suggests that the structure distortion happens between the two states[14–17]. This viscosity, together with the polarity dependence, are reminiscent of the twisted intramolecular charge-transfer model, which has been observed in 4-(dimethylamino)benzonitrile derivatives and other molecules[14–17].

To further probe the structure distortion under excitation, high-pressure PL measurements were conducted on the Au$_{24}$. Au$_{24}$ nanoclusters were loaded, along with toluene as a pressure medium, in a diamond anvil cell and compressed to 3.6 GPa (Fig. 4a). As shown in Fig. 4b, the PL I significantly increases and

blue-shifts under high pressure, and the PL II becomes negligible (note: the PL detector used in the high-pressure study is limited to 1000 nm). This result can be explained if the photo-induced dynamic structure distortion is significantly hindered by the increase of the medium viscosity[33] due to the high pressure, leading to the suppression of the excited-state transition. Our results also indicate that Au$_{24}$ nanocluster can be applied as ratiometric pressure sensor.

Stark measurements were conducted to understand charge-transfer characteristic in Au$_{24}$. Stark spectroscopy is a well-known method to reveal the charge-transfer mechanism[34–36]. Stark measurements provide information regarding intensity change (zeroth derivative component), change of polarizability (first derivative component), and change of dipole moment (second derivative component). For the electrofluorescence (EF) measurement (Fig. 5), excitation was performed at the wavelength where the electric field-induced change in absorption intensity was negligible (Supplementary Fig. 6a–c). From the EF measurement, the change of dipole moment (second) for the PL I of Au$_{24}$ is found to be 1.07 D, indicating the partial charge-transfer occurrence at PL I, which is consistent with the solvent polarity dependence. As the HOMO and LUMO electrons are highly delocalized within the Au$_8$ bi-tetrahedral core in Au$_{24}$[30], this partial charge transfer is better to be termed as electron redistribution, just as other conjugated molecules[37]. Unfortunately, our Stark spectra is limited up to 850 nm, thus we are unable to verify the electron-redistribution character of PL II (~1050 nm).

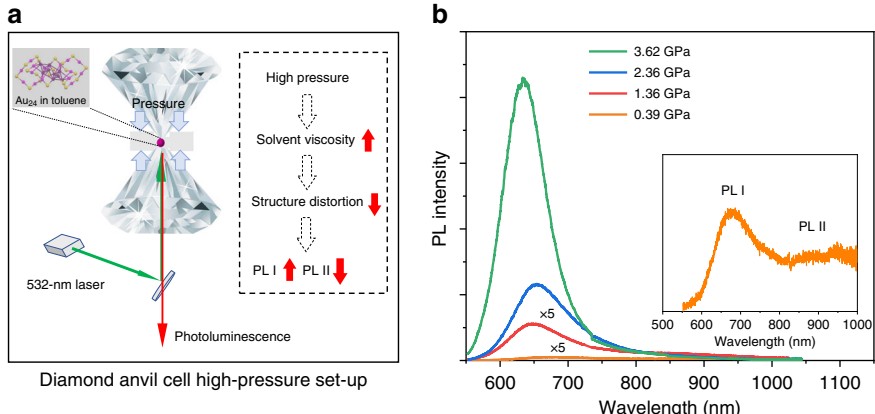

**Fig. 4 Pressure-dependent PL study of Au₂₄. a** The diamond anvil cell high-pressure set-up. $Au_{24}$ in toluene is loaded inside the diamond anvil cell and compressed between the diamond anvils. Inset: mechanism of the pressure-induced change of the dual emission. **b** The pressure-dependent PL spectra of $Au_{24}$. Inset: enlarged PL spectrum of Au24 at 0.39 GPa. PL photoluminescence.

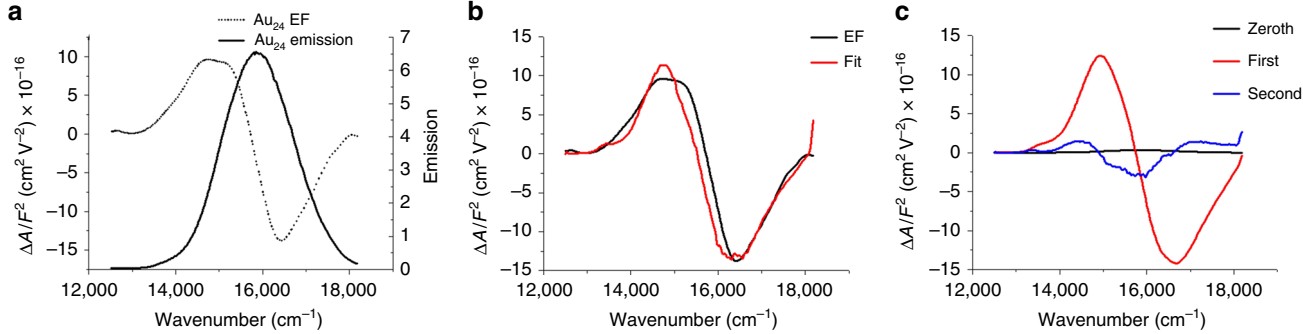

**Fig. 5 Stark spectroscopic measurements on Au₂₄. a** Emission (solid line) and electrofluorescence (EF; dashed). **b** EF (black) and fit (red, the details of the fitting are described in "Methods"). **c** EF fit broken into zeroth, first, and second derivatives.

**Bi-tetrahedral series of Au nanoclusters with dual emission**. In order to gain more structural insight of this nanosecond Au core distortion process, another bi-tetrahedral nanocluster, $Au_{14}Cd(S-Adm)_{12}$, was also studied ($Au_{14}Cd$ for short hereafter). Figure 6 displays the atomic structure, steady-state optical spectra, and excited-state dynamics of $Au_{14}Cd$ (optical data measured in DCM). Anatomy of the atomic structure of $Au_{14}Cd$ is shown in Fig. 6a. The $Au_{14}Cd$ also possesses a bi-tetrahedral $Au_5Cd$ core, but each tetrahedron shares one edge with the other. This bi-tetrahedral $Au_5Cd$ core is further protected by two $Au_4S_5$ motifs and one $AuS_2$ motif. The UV-Vis absorption spectrum is shown in Fig. 6b (purple); a peak at 550 nm with a shoulder at 420 nm can be observed. Based on the previous DFT calculations[31], this 550 nm peak can be mainly attributed to the HOMO–LUMO transition. Both the HOMO and LUMO are distributed around the bi-tetrahedral $Au_5Cd$ core and thus the HOMO–LUMO transition mainly occurs within the $Au_5Cd$ core[31]. The PL spectrum of $Au_{14}Cd$ (Fig. 6b, red) also shows dual emission, with one PL peak at 770 nm and the other at 800–900 nm. The PL excitation spectrum of $Au_{14}Cd$ is similar as the UV-Vis absorption spectrum, and the main peak at 550 nm can be identified (Supplementary Fig. 7). As the two PL both originate from the core-based HOMO–LUMO transition, there is no obvious difference between the PL excitation spectra corresponding to PL at 770 and 900 nm.

The PL dynamics was monitored at different wavelengths by the TCSPC technique. As shown in Fig. 6c, two lifetimes are needed to fit the decay at 850 nm: a short one of 334 ps (93.8%) and a longer component of 86.6 ns (6.2%). When detected at shorter wavelengths (e.g., 680 nm, see Supplementary Fig. 8), the PL

decays faster with increasing amplitude of the short sub-nanosecond component. To further unravel the excited-state dynamics of $Au_{14}Cd$, fs-TA measurements were also conducted. As shown in Fig. 6d, e (TA spectra at typical time delays are shown in Supplementary Fig. 9a–c), after being pumped at 550 nm, a GSB at ~560 nm (corresponding to the HOMO–LUMO transition) can be immediately observed. Multiple ESAs can be identified and their time-dependent evolutions are significantly different. From the first ~100 fs to ~1 ps, there is a decay of ESA at ~730 nm accompanied by the rise of ESA centered at ~670 nm (Fig. 6d, e and Supplementary Fig. 9a). The next process begins from 1 to 10 ps, it can be observed that a decrease of ESA centered at ~630 nm is accompanied by a rise of the ESA at 700–950 nm (Fig. 6d, e and Supplementary Fig. 9b), which further decays to a new ESA from 650 to 850 nm from 10 ps to 400 ps. Because of the significant overlap of the two ESAs, a clear decay can only be observed from 870 to 950 nm. (Fig. 6d, e and Supplementary Fig. 9c) The abnormal increase of the GSB at ~560 nm (the black curve in Fig. 6e) during this time period can be ascribed to a ESA decay located at <600 nm, which overlaps with GSB. Unfortunately, our current probe is limited to >500 nm, and no further information on this ESA can be obtained. The last process (~150 ps) is consistent with the ~200–300 ps component from the TCSPC measurements (Fig. 6c).

We studied the effects of different solvents on the optical properties of $Au_{14}Cd$. The PL spectrum of $Au_{14}Cd$ in hexane is shown in Supplementary Fig. 10 in which the second 800–900 nm peak becomes negligible. The transient absorption for $Au_{14}Cd$ in hexane was also measured (Supplementary Fig. 11a, b). In contrast to $Au_{14}Cd$ in DCM, the electron relaxation to the

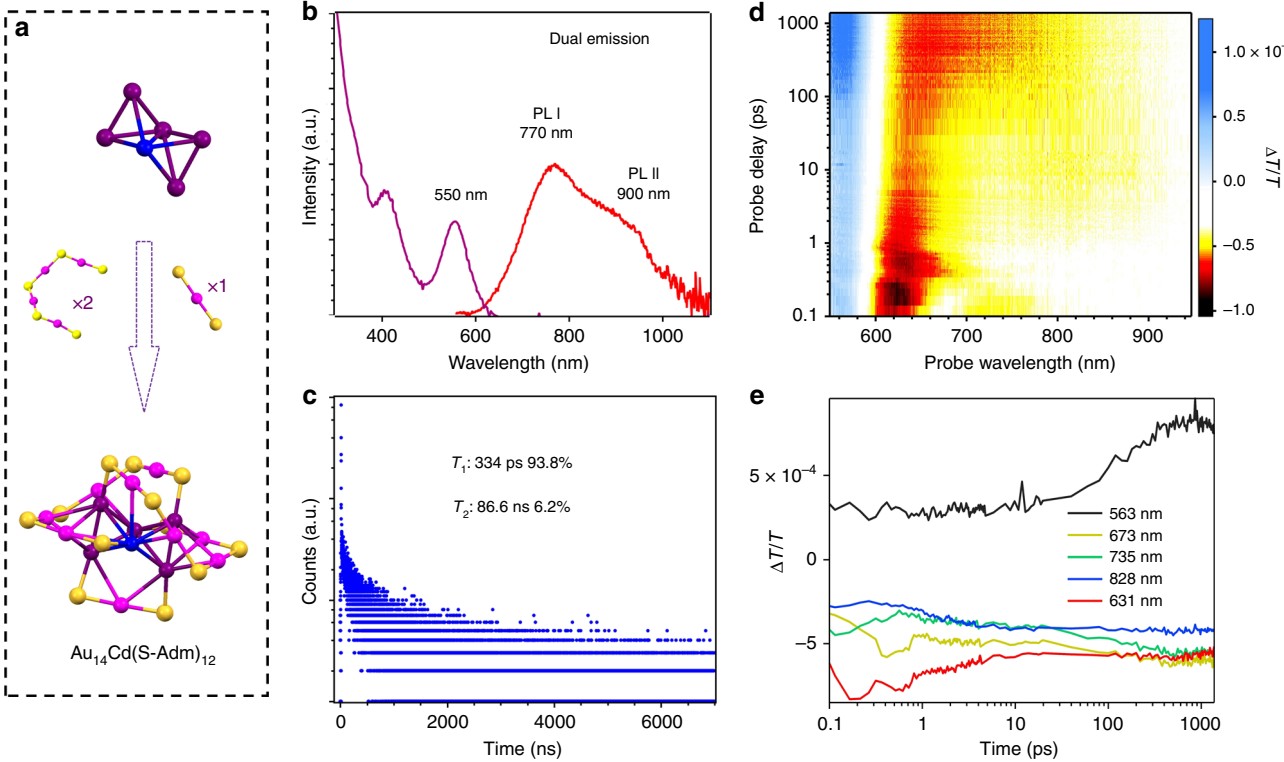

**Fig. 6 Optical properties and excited-state dynamics of Au₁₄Cd. a** Anatomy of the atomic structure of $Au_{14}Cd$[31]. Purple = core Au; magenta = surface Au; blue = Cd; yellow = S; Carbon tails are omitted for clarity. (**b**) UV-Vis absorption (purple) and PL (red) spectra. PL photoluminescence. **c** TCSPC trajectory, which is detected at 850 nm. τ PL lifetime component. **d** fs-TA spectra of Au₁₄Cd pumped at 550 nm, showing the $\Delta T/T$ at all time delays between 550 and 950 nm. **e** Selected kinetic traces at typical wavelengths.

distorted electron-redistribution state has a slower transfer rate of ~500 ps (Supplementary Fig. 11b). In addition, a faster decay back to the ground state after ~900 ps was observed. The TA data are consistent with the PL and TCSPC results. For Au₁₄Cd in hexane, owing to the slower state-transfer rate, a smaller fraction of electrons relaxed to the second distorted electron-redistribution state. This leads to the negligible PL II signal (Supplementary Fig. 10), and the faster decay of PL I (Supplementary Fig. 11b) to ground state is observed. Thus, when Au₁₄Cd is dissolved in a lower dipole solvent, less electron redistribution to the distorted state occurs. This result suggests that similar electron-redistribution mechanisms lead to dual emission in Au₁₄Cd and Au₂₄.

Thus similar relaxation diagrams for Au₁₄Cd and Au₂₄ are obtained from TA and TCSPC measurements. As shown in Fig. 7a, for Au₂₄, the excited electron first relaxes into the electron-redistribution state, which gives birth to the PL I at ~650 nm. Through the next ~ns process, the electron further relaxes into the distorted electron-redistribution state that results in the PL II at ~1050 nm. For the Au₁₄Cd (Fig. 7b), the excited electron arrives at a similar electron-redistribution state, which gives birth to the PL I at ~770 nm. During the next ~200 ps, the electron further relaxes into the distorted electron-redistribution state, which is correlated to the PL II at ~800–900 nm. Compared with the Au₂₄, the gap between the two states is smaller in Au₁₄Cd (0.24 vs 0.68 eV), which is consistent with a faster transition rate (~300 ps vs ~1.6 ns). The smaller energy gap and faster transition between two states suggest that the structural difference of the two excited states in Au₁₄Cd is much smaller than in Au₂₄.

Other Au nanoclusters with bi-tetrahedral core structures have also been studied to further understand the mechanism of structural distortion and electron redistribution. Figure 7c

displays another 24-gold-atom nanocluster protected by 20 phenylethalenethiol (S-PET) ligands (abbreviated as Au₂₄' hereafter). The atomic structure of Au₂₄' was determined by the combination of power XRD and simulations, which shows a different cross-joint bi-tetrahedral $Au_8$ core protected by two $Au_3S_4$ and $Au_5S_6$ surface motifs[32]. It should be noted that this structure has also been verified in a $Au_{24}(SeR)_{20}$ nanocluster with 20 benzeneselenol as surface-protecting ligands[38]. As the selenium atom strongly interferes the excited-state processes through the heavy element effect, here we only discuss the optical properties of the PET-protected Au₂₄'. It can be observed that dual emission (815 nm and 970 nm) was also observed in the Au₂₄' (Fig. 7c) and the state transition was determined to proceed in ~700 ps by the TCSPC and fs-TA measurement (Supplementary Fig. 12 and 13a–c).

We revisit the structural details of the bi-tetrahedral nanoclusters. In Au₁₄Cd, the two tetrahedra shares an edge (Fig. 7b), and this limits the free rotation of individual tetrahedra, thus the structural difference between the first electron-redistribution state and the second distorted electron-redistribution state is much smaller. In contrast, the connection and bonding between the two tetrahedra in Au₂₄ are relatively weaker, and the two tetrahedra have more freedom to move under photoexcitation (Fig. 7a), which induces more significant differences between the two states. In addition, we found that, if there are several (>2) tetrahedra existent in the core of the nanocluster (e.g., the mono-cuboctahedral series[39] and other larger face-centered cubic (fcc) Au nanoclusters[40], Supplementary Fig. 14), the movement would be fully suppressed as every tetrahedron shares several edges and vertexes with other tetrahedrons (i.e., an interlocked kernel structure). Thus photo-induced structural distortion cannot be observed in the mono-cuboctahedral series[39] and other fcc Au

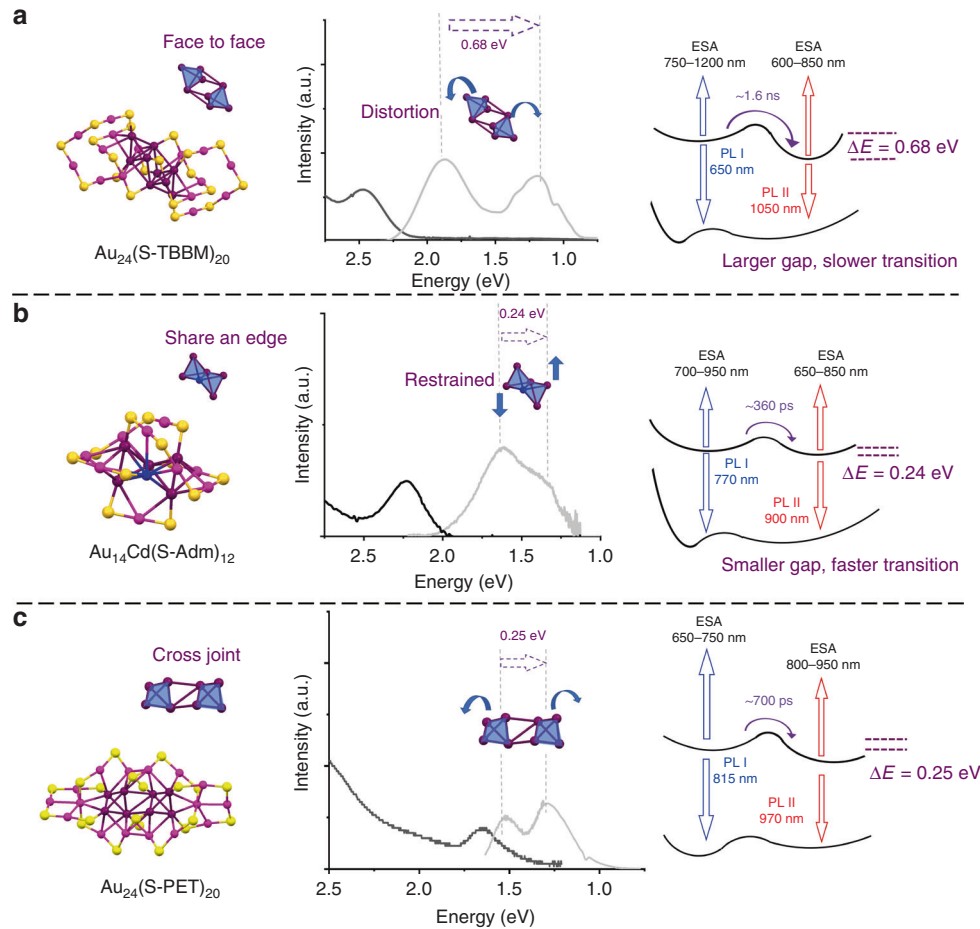

**Fig. 7 Comparison of the three bi-tetrahedral series of Au nanoclusters.** Atomic structure, optical properties, and relaxation diagram of **a** Au$_{24}$, **b** Au$_{14}$Cd, and **c** Au$_{24}$'. In each panel, the atomic structure of the nanocluster is shown on the left, purple = core Au; magenta = surface Au; blue = Cd; yellow = S; the absorption (black) and PL (gray) spectra in the middle; and relaxation diagram on the right. Arrows denote the transitions between different electronic states. ESA excited state absorption.

nanoclusters[40], even if tetrahedra exist in their structures. On the other hand, the core electrons in fcc nanoclusters with several (>2) tetrahedra are more delocalized than in the bi-tetrahedral series. This explains that the charge-transfer characteristic (electron redistribution) can not be identified in fcc nanoclusters by the Stark spectroscopy[39]. The dependence of core structure indicates that the electron redistribution mainly take place within the metal(0) core in bi-tetrahedral nanoclusters, which is different from the metal–ligand charge-transfer mechanism in traditional Au(I) and Ag(I) complex[41,42]. It should be noted that, owing to the ultrasmall size, the core atoms in Au nanoclusters are exposed to the outer environment in some directions (Supplementary Fig. 15), thus they can also interact with the solvents. It also should be noted that the photo-induced structure distortion can be treated as excited-state structural isomerization, thus one would question if there are already isomers in the samples. This possibility has been excluded by using high-purity samples that are re-dissolved from their single crystals, and these single crystals are examined by single-crystal XRD to confirm that no isomers exist inside.

## Discussion
In summary, structural distortion and electron redistribution is unambiguously identified in photo-excited metal nanoclusters, and the atomic-scale structural origin of such excited-state transformation has been fully unraveled. This unexpected

excited-state process in metal nanoclusters induces controllable dual emission, which exhibit highly sensitive and ratiometric responses to multiple external parameters, which makes these nanoclusters quite promising as next-generation probes and sensors. Overall, this work not only provides a simple but versatile strategy to achieve the self-calibrated luminescence responses to multiple environmental parameters from nanoclusters but also presents a paradigm in how to combine the merits of different scales of substances for the design of new materials with synergetic properties. We envision that the strategy and conception developed in this work will not only boost the research of both metal and other semiconductor nanoclusters but also advance the rational design of next-generation luminescent nanosensors, probes, and switches.

## Methods
**Sample preparation**. The synthesis of Au$_{24}$, Au$_{14}$Cd, and Au$_{24}$' follow the published methods[30–32].

**Optical measurements**. The UV–Vis absorption spectra were recorded using a Hewlett Packard 8543 diode array spectrophotometer. The PL spectra were recorded using a FS-5 fluorimeter from Edinburgh Instrument connected with two detectors from 200 to 870 nm and from 850 to 1600 nm and a Horiba Nanolog Hybrid Fluorimeter connected with an Ocean Optics 65000FL spectrograph/charge-coupled device (CCD) (400–1100 nm), as well as a QM 40 spectrophotometer with a InGaAs (500–1700 nm) detector. The nanocluster solution is kept at ~0.1 optical density at the excitation wavelengths (500 nm for Au$_{24}$, 550 nm for Au$_{14}$Cd, and 750 nm for Au$_{24}$') when measuring the PL spectra. The PL

lifetimes were measured by a TCSPC technique with a femtosecond laser (515 nm) as the excitation source. TCSPC trajectories were fitted with bi-exponential functions. Temperature-dependent PL measurements were carried out on a Fluorolog-3 spectrofluorometer (Horiba Jobin Yvon) coupled with an Optistat DN cryostat (Oxford Instruments), an ITC temperature controller, and a pressure gauge. This homo-assembled system allowed us to conduct the temperature-dependent PL experiments from 298 to 80 K. The Au nanoclusters were dissolved in 2-methyltetrohydrofuran for temperature-dependent PL measurements.

**Ultrafast spectroscopy**. For fs-TA spectroscopy, the fundamental output from Yb:KGW laser (1030 nm, 220 fs Gaussian fit, 100 kHz, Light Conversion Ltd) was separated into two light beams. One was introduced to NOPA (ORPHEUS-N, Light Conversion Ltd) to produce a certain wavelength for the pump beam (here we use 520 nm for the $Au_{24}$, 550 nm for $Au_{14}Cd$, and 750 nm for the $Au_{24}'$), the other was focused onto a YAG plate to generate white light continuum as the probe beam. A 1030-nm laser was used to generate visible and NIR probe light, therefore the scattering region between 900 and 1050 nm in the data map is removed. The experiment set-up for the probes at 500–900 nm and 1050–1300 nm were different and used different pump power and pump laser beam size. Therefore, corrections are made to the pump fluences in the reported data. The pump and probe overlapped on the sample at a small angle <10°. The transmitted probe light from sample was collected by a linear CCD array. DCM or hexane solutions of nanoclusters in 1 mm path length cuvettes were excited by the pump. Transient differential transmission signals were obtained by the equation shown below:

$$\frac{\Delta T}{T} = \frac{T(\text{pump-on}) - T(\text{pump-off})}{T(\text{pump-off})} \quad (1)$$

**Stark spectroscopy and data analysis**. Electroabsorption (EA) and EF measurements were taken using a home-built Stark spectrometer[43,44]. White light from a 150 W Xe lamp (Oriel) is focused into the entrance slit of an Acton monochromator, which is equipped with a grating blazed at 500 nm with 1200 groove $mm^{-1}$. The inverse linear dispersion is 5 nm $mm^{-1}$. EA spectra were obtained by focusing the transmitted light onto a photodiode (UDT Sensors) working in photovoltaic mode. The signal was passed through an operational amplifier and measured using a lock-in amplifier (Stanford Instruments SR550) phased with the frequency of the oscillating electric field (441 Hz) applied to the sample and set to acquire at the second harmonic of the signal. To obtain the EF spectra, luminescence from the sample is collected by an emission monochromator (Acton, 600 groove $mm^{-1}$ blazed at 500 nm, spectral resolution 5 nm) and detected by a water-cooled photomultiplier tube (Hamamatsu R928). The electric signal was pre-amplified before arriving at the lock-in amplifier. The signal was collected at twice the frequency of the oscillating electric field, 75 Hz. The Stark effect on absorption or field-induced change in absorption (EA) at wavenumber $\overline{\nu}$, $\Delta A(\overline{\nu})$, were fit by a linear combination of zeroth, first, and second derivatives of absorption spectra as follows:

$$\Delta A(\overline{\nu}) = F^2 \left\{ A_x A(\overline{\nu}) + \frac{B_x}{15hc} \overline{\nu} \frac{d}{d\overline{\nu}} \left( \frac{A(\overline{\nu})}{\overline{\nu}} \right) + \frac{C_x}{30h^2c^2} \overline{\nu} \frac{d^2}{d\overline{\nu}^2} \left( \frac{A(\overline{\nu})}{\overline{\nu}} \right) \right\} \quad (2)$$

Similarly, for the field-induced change in fluorescence (EF):

$$\Delta I(\overline{\nu}) = F^2 \left\{ A_x' I(\overline{\nu}) + \frac{B_x'}{15hc} \overline{\nu}^3 \frac{d}{d\overline{\nu}} \left( \frac{I(\overline{\nu})}{\overline{\nu}^3} \right) + \frac{C_x'}{30h^2c^2} \overline{\nu}^3 \frac{d^2}{d\overline{\nu}^2} \left( \frac{I(\overline{\nu})}{\overline{\nu}^3} \right) \right\} \quad (3)$$

The EA and EF spectra of $Au_{24}$ are collected in a low-temperature glass at 77 K. The zeroth coefficients $A_x$ and $A_x'$ denote the change in transition moment induced by electric field and the field-induced change in emission intensity, respectively. The first derivative coefficient $B_x \cdot (B_x')$ and second derivative coefficient $C_x \cdot (C_x')$ denote the spectral shift and spectral broadening of the EA (EF) spectra. These are correlated with the change in polarizability ($\Delta\alpha$) and dipole moment change ($|\Delta\mu|$), respectively, in the excited state with respect to the ground state.

**High-pressure PL spectroscopy**. $Au_{24}$ in toluene was loaded in a diamond anvil cell high-pressure chamber. Two identical diamond anvils with a culet of 500 µm were employed to generate pressure. A stainless-steel gasket was pre-indented to 400 µm with a drilled hole 200 µm in diameter serving as the sample chamber. The pressure was calibrated using the pressure-dependent ruby fluorescent technique. PL spectra were measured in a Horiba XploRA+ Confocal Raman set-up using the ×10 objectives. Samples were excited by the 532 nm laser and 600 groove $mm^{-1}$ diffraction grating were adopted.

## Data availability

The source data that support the plots within this paper and other finding of this study are available from the corresponding authors upon request.

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

## Acknowledgements

This work was financially supported by the National Science Foundation (DMR-1808675 and DMR- 2002936/2002891). We thank Ms. Jingchun Huang for the preliminary PL measurements and Dr. Meng Zhou for the preliminary TA measurements. This research used resources of the Center for Functional Nanomaterials, which is a U.S. DOE Office of Science Facility at Brookhaven National Laboratory under Contract No. DE-SC0012704.

## Author contributions

Q.L. conceived the project and designed the research advised by R.J. and X.W.G. Q.L. and D.Z. performed the steady-state and TCSPC experiments with help from H.Z., T.C., O.C., M.C., and M.L. D.Z. and H.Z. performed the fs-TA experiments and analyzed the fs-TA data. W.Y.S. and L.A.P. conducted the Stark measurements. Q.L. and X.W.G. conducted high-pressure measurements. Q.L. and J.C. synthesized the samples. Q.L., D.Z., H.Z., R.J. and X.W.G wrote the paper, and all authors commented on it.

## Competing interests

The authors declare no competing interests.
