## [Peer Review File · Nature Communications]

Reviewers' comments:

Reviewer #1 (Remarks to the Author):

Photo-induced processes inducing structural change and/or electron motion or transfer play key roles in many areas of chemistry.

These processes have been identified and evidenced for some organic molecules or complexes (push-pull, Molecular Butterfly) but the bridge to larger systems has not been gap yet. However, for larger materials, e.g. metal nanoparticles, these processes are unexpected because of increased structural rigidity and enhanced electron delocalization.

Due to a strong quantum confinement, nanoclusters, in particular liganded quantum metal clusters behave somewhat like small molecules. Furthermore the staple motifs that protect the metal core present some flexibility towards structural changes. Thus these nanoclusters might be good candidates for evidencing photo-induced processes inducing structural change and/or electron motion or transfer. This is the hypothesis and the aim of the present work.

I do admit that the arsenal of experimental investigations reported in this work is quite impressive, time-resolved fluorescence, excitation state dynamics in the sub-ns regime and in the IR window, solvent, temperature, pressure dependent optical experiments, rational design of cluster structures,...

I thus consider the manuscript to be novel and important, and deserving of publication in Nature Communications. However, I find some of the authors' assumptions to be unjustified and some of their conclusions unconvincing. So I think the paper should be revised before considering it for publication.

In figure 2A and 2B, I can clearly identify at least three states (~600 nm PLI, ~800 nm PLIbis and ~1070nm PLII). Of note the part in ESA dynamics between 900 and 1050 nm is not given. Also I can see some "structurations" in the IR PLII band that might indicate the presence of other excited states. The authors however only consider PLI and PLII contributions.

Their argumentation proving that there is a direct electron-transfer from PL I state to PL II state is based on the "rise" and "decay" of PLII and PLI states. Honestly, I have difficulty to see a rise in Figure 1F. Indeed, if I remove the first 12 points (fs-laser irradiation convoluted by the detection response), what I see is a clear decay of the PLI contribution but a plateau (considering the dispersion of intensity point-by-point) of the PLII contribution vs time. This is not surprising because the time scale of decay of PLII is >66 ns and in the time scale 0 to 2 ns, this decay will appear as a plateau.

However, I am more convinced by the data presented in Fig. 2C, where decay and rise can be observed. However photoinduced structural changes is one of the fundamental excited state dynamic processes, and yet often very challenging to distinguish from competing electronic excited-state relaxation channels having similar timescales.

Now if I assume that this cluster following the HOMO-LUMO transition (GSB) can undergo relaxation towards different channels independently (we have to keep in mind that the landscape for relaxation is tremendously complex and rich), the evolution of the DT/T curves can be simply explained by the competition between GSB->PLI->PLIbis and GSB->PLII. In this scheme PLI should be depleted and PLII should be more populated as observed experimentally.

Another possibility would be the presence in the sample solution of different structural isomers (clusters with slightly different shapes or perhaps the possibility to have the same shape but with different oxidation state for the metal core (core isomerism and staple isomerism)). These two or more different isomers may display different radiation relaxation conducting to both PLI and PLII. Of note there are more and more evidence that isomerism play a significant role in the nanocluster field (thanks for instance to recent theoretical works by Yi Gao et al and ion mobility measurements by Pradeep et al and Antoine et al.). It could be important to rule out this possibility for the present study.

Solvent dependent experiments and pressure dependent experiments are interesting, but do they prove anything concerning a direct electron-transfer from PL I state to PL II state ? I am not so sure. Indeed, the solubility of Au₂₄ in hexane is very low, meaning that Au₂₄ in such solvent will rapidly self-aggregate. And the solvent dependent measurements are more indicative of a transition from Au₂₄ monomeric units to Au₂₄ oligomeric units that present aggregation induced emission. Look for instance to the impressive similarity between spectra in hexane and butanol and the one reported in solid state (where aggregated species leads to this AEI effect). The pressure dependent experiments can also be explained by a transition from isolated Au₂₄ to "bulk-like" Au₂₄ under pressure.

I agree that solvent effects (both polarity and viscosity) on the photoinduced structural change of organic molecules have been reported and play a significant role. However in the case of such molecule (and push-pull or butterfly molecules are nice examples), there is a direct interaction between the solvent molecules and the "delocalized electron" system. Furthermore the solvent effect has found to be dependent on peculiar electronic excited states with charge transfer character. In the present work, the gold core (bi-tetrahedra Au₈ core) is protected by staple motifs that inhibit a strong interaction of the gold core with solvent molecules. Furthermore, it was mentioned that from HOMO LUMO orbitals, electrons are highly delocalized within the Au₈ bi-tetrahedral core in Au₂₄, limiting charge transfer character of transitions.

The Stark spectroscopy measurements permit to evaluate the change of dipole moment (2nd) for the PL I of Au₂₄ to be 1.07 D. This change might have a different origin. The HOMO-LUMO transition is mainly occurring within the metal core and the relaxation towards the state PLI may be or "surface state" nature where some ligand-to-core charge transfer may explain this variation in dipole moment. I would be more convincing if different changes in dipole moment were reported with a solvent with different polarity.

Other comments.

Fig. 1C reports PL excitation measurements at 1050 nm, while PL decay in Figure 1E are collected between 900 and 1000 nm. Would better to report data collected from 900 to 1300 nm.

It could be indicative to add the response function of the detector in Fig. 1F (to see which point needs to be considered at the beginning).

Fig; 2C and 6C. I am surprised about the behavior of the GSB evolution as a function of time. Pump occurs at 520 nm and probe at similar wavelength, Thus considering the lifetimes observed 1.6 ns and 330 ps, I would have expected to see depletion at long pump-probe time (~1ns). The contrary is observed and I don't understand their explanation.

Table S1 : please add values for polarity and viscosity for the used solvents.

Reviewer #2 (Remarks to the Author):

The authors report on the dynamics of dual emission from Au cluster that are terminated by different Au-organic complexes. The origin of the dual emission is attributed to a structural distortion of the Au cluster core.

I can understand the optical characterization and the interpretation of a two channel model (PL I and PL II) for the photoluminescence decay, where the lower energy recombination (PL I) is in competition with the high energy one (PL II). In that sense it acts as a non-radiative decay mechanism for that particular channel (being then radiative in the second one).

I do not see the unambiguous proof of the structural distortion. I would regard the system as a core-shell particle (with a soft outer shell composed by the Au-organic complexes) that exhibits emission from both core and shell with different rates. In that view it would be reasonable that the shell transition rate depends strongly on the environment. In a different vocabulary the PL II could be called radiative defect emission that is sensitive to the environment.

If there were a distortion, what about vibrational modes? Shouldn't such a distortion be oscillating, thus be an acoustic vibrational mode? Here Raman spectroscopy on this system would be very interesting.

And the paper does not report any "dark " characterization, thus the non-distorted case (if there is a photo-induced distortion) is not probed at all.

The system should be small enough for ab initio modeling. Can such data be provided?

Also the claim of electron redistribution should be supported by theory. In the picture of the 2 decay channels no pronounced redistribution is needed, and one would look at the process in terms of exciton relaxation.

These doubts on the origin don't affect the usefulness of the system for sensing, and the interesting optical data.

Therefore I recommend major revisions.

Response to reviewers' comments

Response to Reviewer 1 (pages 1-10)

Response to Reviewer 2 (pages 11-19)

Our response is in **blue** and revisions are in **red**.

Reviewer #1 (Remarks to the Author):

Photo-induced processes inducing structural change and/or electron motion or transfer play key roles in many areas of chemistry. These processes have been identified and evidenced for some organic molecules or complexes (push-pull, Molecular Butterfly) but the bridge to larger systems has not been gap yet. However, for larger materials, e.g. metal nanoparticles, these processes are unexpected because of increased structural rigidity and enhanced electron delocalization. Due to a strong quantum confinement, nanoclusters, in particular liganded quantum metal clusters behave somewhat like small molecules. Furthermore, the staple motifs that protect the metal core present some flexibility towards structural changes. Thus, these nanoclusters might be good candidates for evidencing photo-induced processes inducing structural change and/or electron motion or transfer. This is the hypothesis and the aim of the present work. I do admit that the arsenal of experimental investigations reported in this work is quite impressive, time-resolved fluorescence, excitation state dynamics in the sub-ns regime and in the IR window, solvent, temperature, pressure dependent optical experiments, rational design of cluster structures,... I thus consider the manuscript to be novel and important, and deserving of publication in Nature Communications. However, I find some of the authors' assumptions to be unjustified and some of their conclusions unconvincing. So I think the paper should be revised before considering it for publication.

Response: Thanks for this comment!

(1) In figure 2A and 2B, I can clearly identify at least three states (~600 nm PLI, ~800 nm PLIbis and ~1070nm PLII). Of note the part in ESA dynamics between 900 and 1050 nm is not given. Also I can see some "structurations" in the IR PLII band that might indicate the presence of other excited states. The authors however only consider PLI and PLII contributions. Their argumentation proving that there is a direct electron-transfer from PL I state to PL II state is based on the "rise" and "decay" of PLII and PLI states. Honestly, I have difficulty to see a rise in Figure 1F. Indeed, if I remove the first 12 points (fs-laser irradiation convoluted by the detection response), what I see is a clear decay of the PLI contribution but a plateau (considering the dispersion of intensity point-by-point) of the PLII contribution vs time. This is not surprising because the time scale of decay of PLII is >66 ns and in the time scale 0 to 2 ns, this decay will appear as a plateau. However, I am more convinced by the data presented in Fig. 2C, where decay and rise can be observed. However, photoinduced structural changes is one of the fundamental excited state dynamic processes, and yet often very challenging to distinguish from competing electronic excited-state relaxation channels having similar timescales. Now if I assume that this cluster following the HOMO-LUMO transition (GSB) can undergo relaxation towards different channels independently (we have to keep in mind that the landscape for relaxation is tremendously complex and rich), the evolution of the DT/T curves can be simply explained by the competition between GSB->PLI->PLIbis and GSB->PLII. In this scheme PLI should be depleted and PLII should be more populated as observed experimentally.

Response: Thank you very much for the insightful comments! We realized the original figures might not be clear enough, thus we have provided updated Figure 1, Figure 2 and Figure S1.

We used 1030 nm to generate visible and near-IR probe light, therefore the scattering region between 900 and 1050 nm in the data map is removed. In the updated Fig.2A and Fig. S1, we plotted these two regions

in one figure and now one can observe that in the near-IR region, it is just one very broad ESA starting from ~ 750 nm and extends to 1200 nm; there is no additional ESA “peak” at 1050 nm. This broad ESA corresponds to the photoluminescence (PL I) ~ 670 nm.

We updated Fig. 1F with plots that show clearly concurrent rise and decay processes for the two PL peaks.

The “structurations” in the NIR PL II ~ 1050 nm is actually from the solvent which was also reported for other different Au nanocluster samples (e.g. Kauffman, D. R. et al., *J. Am. Chem. Soc.*, 2012, 134, 10237-10243, see Fig S3 in https://pubs.acs.org/doi/suppl/10.1021/ja303259q/suppl_file/ja303259q_si_001.pdf). We have noted this solvent “structure” in the updated Figure 1B with a *. Thus, there are only two PL states for the Au_{24} .

We agree with reviewer that the excited state evolution is quite complex and rich. Based on the dynamic interchange observed in both PL and TA studies, we believe that our two-channel model (new Figure 2C) is reasonable and can describe the results well.

Revisions: Updated Figure 1 and Figure 2, and Figure S1 added.

Figure 1. Optical properties and PL dynamics of Au_{24} . (A) Anatomy of the atomic structure of Au_{24} determined by single-crystal X-ray diffraction³⁰. Purple=Core Au; Magenta=Surface Au; Yellow=S; Carbon tails are omitted for clarity. (B) UV-Vis absorption (black) and PL (grey) spectra (the * in B denotes the spectrum “structuration” which is induced by the solvent). (C) PL excitation spectra for the two emissions, measured at 650 nm (blue) and 1050 nm (red), respectively. (D) and (E) Time-correlated-single-photon-counting (TCSPC) trajectories of the PL I (detected from 550 to 750 nm) and PL II (detected from 900 to 1000 nm). (F) Comparison of the TCSPC trajectories of PL I (blue) and PL II (red) in the early 1.5 ns.

Figure 2. Excited-state dynamics of Au₂₄: Direct identification of the transition between the two states. (A) Map of the TA spectra of Au₂₄ pumped at 520 nm which shows the $\Delta T/T$ at all time-delays between 500–900 nm and 1050–1300 nm. (B) Selected kinetic traces at typical wavelengths. (C) Relaxation diagrams for Au₂₄.

Figure S1. fs-TA spectra of Au₂₄ at selected time delays (solvent: DCM).

(2) Another possibility would be the presence in the sample solution of different structural isomers (clusters with slightly different shapes or perhaps the possibility to have the same shape but with different oxidation state for the metal core (core isomerism and staple isomerism)). These two or more different isomers may display different radiation relaxation conducting to both PLI and PLII. Of note there are more and more evidence that isomerism play a significant role in the nanocluster field (thanks for instance to recent

theoretical works by Yi Gao et al and ion mobility measurements by Pradeep et al and Antoine et al.). It could be important to rule out this possibility for the present study.

Response: Thanks for the comments. We agree that the isomers might be one possibility, thus we have carefully excluded this possibility during our experiment by the following two ways. First, our Au₂₄ and Au₁₄Cd samples were re-dissolved from their single crystals, and these single crystals were examined by the single-crystal X-ray diffraction which definitely had no isomers inside. Second, we monitored the absorption of samples in every solution before and after our time-resolved optical measurement. It should be noted that isomers show different absorption which can be easily distinguished by UV-Vis measurement (Tian *et al. Nature Communications* **6**, 2015, 8667, <https://www.nature.com/articles/ncomms9667> ; Chen *et al. J. Am. Chem. Soc.* **2016**, *138*, 5, 1482-1485, <https://pubs.acs.org/doi/10.1021/jacs.5b12094>), and this is not observed in our measurements.

Revision: New comments added in page 13

“It also should be noted that the photo-induced structure distortion can be treated as excited-state “structure isomerization”, thus one would question if there are already isomers in the samples. This possibility has been excluded by using “high-purity” samples which are re-dissolved from their single crystals, and these single crystals are examined by single-crystal X-ray diffraction to confirm that no isomers exist inside.”

(3) Solvent dependent experiments and pressure dependent experiments are interesting, but do they prove anything concerning a direct electron-transfer from PL I state to PL II state ? I am not so sure. Indeed, the solubility of Au₂₄ in hexane is very low, meaning that Au₂₄ in such solvent will rapidly self-aggregate. And the solvent dependent measurements are more indicative of a transition from Au₂₄ monomeric units to Au₂₄ oligomeric units that present aggregation induced emission. Look for instance to the impressive similarity between spectra in hexane and butanol and the one reported in solid state (where aggregated species leads to this AEI effect). The pressure dependent experiments can also be explained by a transition from isolated Au₂₄ to “bulk-like” Au₂₄ under pressure. I agree that solvent effects (both polarity and viscosity) on the photoinduced structural change of organic molecules have been reported and play a significant role. However in the case of such molecule (and push-pull or butterfly molecules are nice examples), there is a direct interaction between the solvent molecules and the “delocalized electron” system. Furthermore the solvent effect has found to be dependent on peculiar electronic excited states with charge transfer character. In the present work, the gold core (bi-tetrahedra Au₈ core) is protected by staple motifs that inhibit a strong interaction of the gold core with solvent molecules. Furthermore, it was mentioned that from HOMO LUMO orbitals, electrons are highly delocalized within the Au₈ bi-tetrahedral core in Au₂₄, limiting charge transfer character of transitions.

Response: Thanks for the comments. More experiments have been conducted to address these issues.

First, we have conducted more solvent-dependent experiments in which Au₂₄ has better solubility. It should be noted that it is not a “sudden switch” from DCM to hexane, but a “smooth transition” from DCM (polarity index 3.1), toluene (polarity index 2.4) to hexane (polarity index 0.1) (updated Figure 3). The nanocluster is well-dissolved in DCM and toluene. Second, the nanoclusters do not form aggregates even in hexane. We show this by drop-casting a Au₂₄ hexane solution onto a TEM grid, and are able to image dispersed clusters. If no aggregate is observed on TEM grids after drying, it is unlikely that aggregates would exist in the original solution, because the drying process would not break the original aggregates. Furthermore, as we already showed in Figure S2, the UV-Vis absorbance of Au₂₄ in hexane is almost the same as in DCM (both ~0.1 OD), this also indicates no appreciable aggregation or solvent-induced other structural changes (e.g. isomerization) in going from DCM to hexane.

Figure for response: TEM image of the Au₂₄ nanoclusters drop-cast from the hexane solution.

In addition, on the “aggregation” issue, we find that the solid-state PL of Au₂₄ crystal (closely packed or aggregated) and the Au₂₄-polymer film (dispersed in matrix) are almost the same. This indicates that aggregation should not be the direct reason for the change of PL properties as long as the cluster structural distortion is hindered in solid states. Thus, we have replaced the “aggregation-state PL” with “solid-state PL” in the revised manuscript.

We realized there is a need to explain more on the high-pressure study. In the experiment, there is toluene as the solvent between the two diamonds. By the increase of the high-pressure, the viscosity of the toluene solvent increases (*International Journal of Thermophysics* **volume 18**, pages367–378(1997) <https://link.springer.com/article/10.1007/BF02575167>) which hinders the excited-state structure-distortion. Thus, the high-pressure study is a more “clean” way to prove the viscosity-dependence of the PL as it doesn’t need to replace the original solvent. This high-pressure method has been applied to support the structure-distortion process in many twisted charge-transfer organic fluorophores,(*J. Chem. Phys.* 104, 9431 (1996) <https://aip.scitation.org/doi/pdf/10.1063/1.471687>) and the PL change of our clusters under the increasing pressure are similar as previously reported twisted charge-transfer molecules. Pressure within 0-3 GPa are unable to “press the clusters together” as the toluene medium becomes more and more viscous which actually prevents the movement of individual nanoclusters. We have updated Figure 4 for readers to understand it more easily.

We understand there is also a need to provide more details on the geometry of the ultrasmall Au nanoclusters and also the detailed concept of the Au kernel and Au-shell in nanoclusters. Generally, we treat those fundamental Au units (e.g. Au₄ tetrahedron, Au₁₃ cuboctahedron) as the Au kernels; while those outside linear Au-S oligomers as the surface motifs.(*Chem. Rev.* 2016, 116, 18, 10346, <https://pubs.acs.org/doi/10.1021/acs.chemrev.5b00703>) However, due to the ultrasmall size, the structures of Au nanoclusters are quite “open” and the Au kernel atoms are actually exposed to the environment and

can directly interact with the solvents (see newly added Figure S15). Thus, it's reasonable that the kernel-based PL is also sensitive to the environment.

Revisions: Updated Figures 3 &4, and Figure S15 added.

Figure 3. Solvent polarity and viscosity dependence, as well as the solid-state PL of Au₂₄. Comparison of the PL (A) and TCSPC (B) of Au₂₄ in DCM (black), toluene (blue) and hexane (red). TCSPC trajectories were detected from 550 to 650 nm. (C) Comparison of the PL of Au₂₄ in DCM (top), butanol (middle) and solid state (crystal or film, bottom). From top to bottom, the overall QY of Au₂₄ increases from 2 % to 30 %, which is accompanied by an increase in the ratio of PL I / PL II and the blue shift of the PL I. Inserted photographs: Au₂₄ in butanol and solid-state film and single crystals under UV lamp.

Figure 4. Pressure-dependent PL study of Au₂₄. (A) The diamond anvil cell high-pressure setup. (B) The pressure-dependent PL spectra of Au₂₄.

Figure S15 The “open” Au₈ kernel of Au₂₄ from different viewing directions.

New comments added in Page 8: This result can be explained if the photo-induced dynamic structure distortion is significantly hindered by the increase of the medium viscosity³³ due to the high pressure, leading to the suppression of the excited-state transition.

New comments added in Page 13: It should be noted that due to the ultrasmall size, the core atoms in Au nanoclusters are exposed to the outer environment in some directions (Figure S15).

(4) The Stark spectroscopy measurements permit to evaluate the change of dipole moment (2nd) for the PL I of Au₂₄ to be 1.07 D. This change might have a different origin. The HOMO-LUMO transition is mainly occurring within the metal core and the relaxation towards the state PLI may be of “surface state” nature where some ligand-to-core charge transfer may explain this variation in dipole moment. I would be more convincing if different changes in dipole moment were reported with a solvent with different polarity.

Response: We thank you for the comments. Trying different solvent polarity to investigate the different dipole moment would be a really good idea. Our collaborator, Peteanu group, has tried to investigate different dipole moment change from solvatochromism, but found that the Stark measurement, which was conducted at low-temperature (liquid nitrogen) using glass-forming solvents, usually doesn't conserve the solvent orientational polarizability (*J. Phys. Chem. C* 2018, 122, 22, 11961; <https://pubs.acs.org/doi/10.1021/acs.jpcc.7b12025>).

Other Comments

(1) Fig. 1C reports PL excitation measurements at 1050 nm, while PL decay in Figure 1E are collected between 900 and 1000 nm. Would better to report data collected from 900 to 1300 nm. It could be indicative to add the response function of the detector in Fig. 1F (to see which point needs to be considered at the beginning).

Response: Thanks for the comment. We have added the new data. The efficiency of our TCSPC detector > 1000 nm is very low so that we can't get good signals in that range. But the updated data (Figure 1F) can clearly distinguish the response function of the detector and the rise of the signal. We have also measured the PLE at 900 nm, which is almost the same as 1050 nm (see below)

Figure for response: the PLE spectrum of Au₂₄ detected at PL 900 nm.

Revision: New updated Figure 1F

(2) Fig; 2C and 6C. I am surprised about the behavior of the GSB evolution as a function of time. Pump occur at 520 nm and probe at similar wavelength, Thus considering the lifetimes observed 1.6 ns and 330 ps, I would have expected to see depletion at long pump-probe time (~ 1 ns). The contrary is observed and I don't understand their explanation.

Response: Thanks for the comments. In TA spectra, as long as the molecule is still in the excited state (not back to ground state yet), the GSB will be there, no matter the molecule is in which excited state. The 1 ns and 300 ps process is a transition process between excited states, from excited-state I to II, without decaying back to the ground state in 1 ns time scale. Therefore, there is a negligible decay of GSB in this time range. The GSB will decay only when the excited molecule comes back to the ground state.

In some cases, the GSB will not decay but increase due to spectral overlap issue. We further conducted a fs-TA of our third bitetrahedral Au NC, the Au₂₄-S-PET whose absorption and GSB is at 750 nm (newly added Figure S13). Because there is another ESA starting from ~ 700 nm which overlaps with the GSB at ~ 750 nm, during $\sim 10 - 1000$ ps the GSB looks “growing” due to the decay of the overlapping ESA. In Au₂₄ and Au₁₄Cd, the GSB is ~ 500 to 550 nm and there could be another ESA < 500 nm which decays at this time-range (~ 1 ns and 300 ps), thus making the GSB look like it is “growing”.

Revision. New Figure S13 added.

Figure S13 (A) TA spectra of Au₂₄ pumped at 520 nm which shows the $\Delta T/T$ at all time-delays between 500–900 nm and 1050–1300 nm. (B) fs-TA spectra of Au₂₄ at typical time delays. (C) Selected kinetic traces at typical wavelengths.

(3) Table S1 please add values for polarity and viscosity for the used solvents.

Response: Table S1 has been revised as suggested.

Revision: Updated Table S1

Solvent	PL I	Polarity Index	Viscosity (cp r.t.)
DCM	670 nm	3.1	0.43
Toluene	650 nm	2.4	0.59
Hexane	610 nm	0.1	0.31
DCM	670 nm	3.1	0.43
1,2-Dichlorobenzene	660 nm	2.7	1.32
1-Butanol	600 nm	3.9	2.95
1-Octanol	590 nm	3.4	10.6

Reviewer #2 (Remarks to the Author):

The authors report on the dynamics of dual emission from Au cluster that are terminated by different Au-organic complexes. The origin of the dual emission is attributed to a structural distortion of the Au cluster core. I can understand the optical characterization and the interpretation of a two channel model (PL I and PL II) for the photoluminescence decay, where the lower energy recombination (PL I) is in competition with the high energy one (PLI). In that sense it acts as a non-radiative decay mechanism for that particular channel (being then radiative in the second one).

(1) I do not see the unambiguous proof of the structural distortion. I would regard the system as a core-shell particle (with a soft outer shell composed by the Au-organic complexes) that exhibits emission from both core and shell with different rates. In that view it would be reasonable that the shell transition rate depends strongly on the environment. In a different vocabulary the PL II could be called radiative defect emission that is sensitive to the environment.

Response: Thanks for the comments. We added more experimental data to support our “structural distortion” model during the transition process from the state I to state II. Viscosity-dependent and high-pressure study is the widely-accepted method to prove the excitation-state structural-distortion in dual emitting materials which have been generally utilized in the study of “twisted charge transfer molecules and fluorescent molecular motors”. (*Chem. Rev.* 2003, 103, 10, 3899-4032 <https://pubs.acs.org/doi/abs/10.1021/cr9407451>; *Chem. Rev.* 1993, 93, 1, 507-535 <https://pubs.acs.org/doi/abs/10.1021/cr00017a022>; *Spectrochimica Acta Part A: Molecular and Biomolecular Spectroscopy*, 222, 2019, 117244. <https://www.sciencedirect.com/science/article/pii/S1386142519306341>). We have added more viscosity-dependent experimental results to confirm our model. Our justification is as follows.

First, by the combination of fs-TA, TCSPC and steady-state results, it has been confirmed that the ~1 ns process (Figure 1F, Figure 2) is a transition process from state PL I to state PL II (see the updated Figure 1 and Figure 2)

Figure 1. Optical properties and PL dynamics of Au₂₄. (A) Anatomy of the atomic structure of Au₂₄ determined by single-crystal X-ray diffraction³⁰. Purple=Core Au; Magenta=Surface Au; Yellow=S; Carbon tails are omitted for clarity. (B) UV-Vis absorption (black) and PL (grey) spectra (the * in B denotes the spectrum “structuration” which is induced by the solvent). (C) PL excitation spectra for the two emissions, measured at 650 nm (blue) and 1050 nm (red), respectively. (D) and (E) Time-correlated-single-photon-counting (TCSPC) trajectories of the PL I (detected from 550 to 750 nm) and PL II (detected from 900 to 1000 nm). (F) Comparison of the TCSPC trajectories of PL I (blue) and PL II (red) in the early 1.5 ns.

Figure 2. Excited-state dynamics of Au₂₄: Direct identification of the transition between the two states. (A) Map of the TA spectra of Au₂₄ pumped at 520 nm which shows the $\Delta T/T$ at all time-delays between 500–900 nm and 1050–1300 nm. (B) Selected kinetic traces at typical wavelengths. (C) Relaxation diagrams for Au₂₄.

Second, we demonstrate that such a ~1 ns state-to-state transition process is directly dependent on the viscosity of the solvent. Besides the previous steady-state viscosity-dependent data, we added new viscosity-dependent TCSPC data. We chose the combination of DCM (viscosity 0.413 cP at 25 °C) and 1,2-dichlorobenzene (viscosity 1.32 cP at 25 °C) and adjusted the viscosity of the mixture by changing their ratios (New Figure S5). It was found that by the increase of 1,2- dichlorobenzene (i.e. VISCOSITY INCREASE) this nanosecond state-to-state transition process is greatly slowed down. Furthermore, if we directly dissolve the Au₂₄ in a highly viscous solvent such as butanol (viscosity 2.59 cP at 25 °C), the state-

to-state transition cannot be detected by TCSPC (New figure S3) and only the single PL I at 600 nm can be observed in the steady-state PL.

Figure S5: TCSPC trajectory of the PL I at 650 nm of Au₂₄ in DCM/1,2-dichlorobenzene with different ratios.

Figure S3: TCSPC trajectory of Au₂₄ in butanol. Inset is the PL of Au₂₄ in butanol, the TCSPC data was detected at 600 nm.

This viscosity-dependence of the state-to-state transition (~1 ns) is a direct proof that such a transition process is related with the movement of atoms in the nanocluster (here we use the word “structure distortion”). Such a method has been reported and widely-accepted in the study of twisted

[charge transfer dual-emitting dyes and those fluorescence molecular motors](https://pubs.acs.org/doi/abs/10.1021/cr9407451) (*Chem. Rev.* 2003, 103, 10, 3899-4032 <https://pubs.acs.org/doi/abs/10.1021/cr9407451> ; *Chem. Rev.* 1993, 93, 1, 507-535 <https://pubs.acs.org/doi/abs/10.1021/cr00017a022>; *Spectrochimica Acta Part A: Molecular and Biomolecular Spectroscopy*, 2019, 222, 117244. <https://www.sciencedirect.com/science/article/pii/S1386142519306341>). This is also the core and aim of this work, that is, ultrasmall nanoclusters show evidence for photo-induced processes (structural changes) which have only been reported in molecules.

We realized there is a need to explain more details on the high-pressure effect. In the experiment, there is toluene as the medium between the two diamonds. By the increase of the high-pressure, the viscosity of the solvent increases (*International Journal of Thermophysics* **volume 18**, pages367–378(1997) <https://link.springer.com/article/10.1007/BF02575167>) which hinders the structure-distortion. Thus, the high-pressure study is another more “clean” way to prove the viscosity-dependence of the PL as it doesn’t need to replace the original solvent. [The high-pressure ratiometric response has been applied to support the structure-distortion process in many structurally-twisted charge-transfer organic fluorophores](https://aip.scitation.org/doi/pdf/10.1063/1.471687) (*J. Chem. Phys.* 104, 9431 (1996) <https://aip.scitation.org/doi/pdf/10.1063/1.471687>; *Chem. Rev.* 2003, 103, 10, 3899-4032 <https://pubs.acs.org/doi/abs/10.1021/cr9407451>). We have updated the Figure 4 for readers to understand this more easily.

Figure 4. Pressure-dependent PL study of Au₂₄. (A) The diamond anvil cell high-pressure setup. (B) The pressure-dependent PL spectra of Au₂₄.

These experiments are generally used to determine the excited-state structure-distortion for the dual-state fluorophores. On the other hand, as far as we know, there is no reported process of exciton decay or transfer (e.g. core to surface) in quantum dots or other larger nanoparticles, for which the core-shell model is appropriate, that show this kind of solvent viscosity dependence. For these reasons, we stand by our explanation of structural distortion.

Revision: Newly added Figure S3 and Figure S5. Newly updated Figures 1,2 and 4 (shown in the above response).

(2) If there were a distortion, what about vibrational modes? Shouldn’t such a distortion be oscillating, thus be an acoustic vibrational mode? Here Raman spectroscopy on this system would be very interesting.

Response: We thank reviewer for raising this interesting question. The structural distortion or transition is triggered by vibrational mode that is coupled to the electron strongly. Usually it is the optical phonon mode that couples to electron strongly. Unfortunately, we don't have this information currently. Sometimes the coherent vibration that couples to electron can show up on TA spectra as oscillation only if 1) the laser pulse is much shorter compared to the vibration period time, 2) the vibration is strongly coupled to a certain optical transition, and 3) the vibration has a long enough coherent time. In this kind of metal clusters, we expect the vibrations are strongly coupled such that they dephase quickly, therefore we cannot see their coherent in-phase effect.

We thank the reviewer for suggesting Raman spectrum which can provide vibration information for the ground state. The excited state vibrational mode would not show up in ground state Raman spectra. Also, unfortunately, the clusters have very strong and broad PL which prevents Raman measurement with conventional 532 nm laser in Raman setup. In addition, both acoustic and optical phonons would be very low frequencies (<200 cm⁻¹), such Raman setups are not popular. A theoretical study is needed to identify the mode coupled to distortion.

(3) And the paper does not report any “dark“ characterization, thus the non-distorted case (if there is a photo-induced distortion) is not probed at all.

Response. Thanks for the comments. We have added more discussions on the “dark” case with no photo-induced distortion and compared their structure and dynamics.

First, as we mentioned above, we added the new TCSPC data of Au₂₄ in viscous solvent 1-butanol (viscosity 2.59 cP at 25 °C) and the mixture of DCM/1,2-dichlorobenzene. It can be observed in newly added Figure S3 that in this viscous butanol environment, no quick decay of the state I can be detected and there is just one PL band observed, indicating the distortion to state II has been totally suppressed. And in the gradual increase of the viscosity of DCM/1,2-dichlorobenzene, the structure-distortion process has been gradually slowed down. All these new data can be treated as “dark cases”, that is, a sample which can “intrinsically distort” but being “hindered” by the outer viscous environment.

Second, we added more discussions on the structure and optical results on other Au NCs which show no excited-state distortion. In these Au nanoclusters— which have more than four interconnected tetrahedral Au₄ units in their Au kernels (newly added Figure S13), the movement of Au₄ tetrahedral unit would be fully suppressed as every tetrahedron shares several edges and vertexes with neighboring Au₄ and being “interlocked” with each other. Thus, these nanoclusters only show one single PL state and in their fs-TA and TCSPC spectra, no state-transition from ~100 ps to 1 ns can be observed as in the current bitetrahedral series (optical data published in our previous work, *J. Am. Chem. Soc.* 2019, 141, 5314-5325, <https://pubs.acs.org/doi/10.1021/jacs.8b13558>; *J. Phys. Chem. Lett.* 2017, 8, 4023-4030, <https://pubs.acs.org/doi/abs/10.1021/acs.jpcclett.7b01597>).

Revision: New Figure S3, S5 and S14 are added. New comments added in page 12.

Figure S5: TCSPC trajectory of the PL I at 650 nm of Au₂₄ in DCM/1,2-dichlorobenzene with different ratios.

Figure S3: TCSPC trajectory of Au₂₄ in butanol. Inset is the PL of Au₂₄ in butanol, the TCSPC data was detected at 600 nm.

Figure S14 Atomic structure of Au₂₁ and Au₂₈ determined by the single-crystal X-ray diffraction.^{1,2} In these two nanoclusters, the four tetrahedra share vertex and sides and thus being interlocked. The “movement” and structure-distortion of the Au₄ units have been hindered.

Revision: new comments added in page 12:

“In addition, we found that, if there are several (>2) tetrahedra existent in the core of the nanocluster (*e.g.* the mono-cuboctahedral series³⁸ and other larger *fcc* Au nanoclusters⁴⁰, Figure S14), the movement would be fully suppressed as every tetrahedron shares several edges and vertexes with other tetrahedrons (*i.e.* an interlocked kernel structure). Thus, photo-induced structural distortion can not be observed in the mono-cuboctahedral series³⁸ and other *fcc* Au nanoclusters,⁴⁰ even if tetrahedra exist in their structures.”

(4) The system should be small enough for ab initio modeling. Can such data be provided? Also the claim of electron redistribution should be supported by theory. In the picture of the 2 decay channels no pronounced redistribution is needed, and one would look at the process in terms of exciton relaxation.

Response: We have asked our collaborators to conduct simulations, but unfortunately the simulations on excited states of heavy metal nanoclusters are very difficult and the results are often significantly different from experiment (by orders of magnitude).

The electron redistribution (*i.e.* charge transfer) is confirmed by the Stark PL measurement (Figure 5) and the solvatochromism. These two methods are widely adopted in the study of fluorescence (*Chem. Rev.* 2003, 103, 10, 3899-4032 <https://pubs.acs.org/doi/abs/10.1021/cr9407451>; *J. Am. Chem. Soc.*, 2004, 126(36): 11154. <https://pubs.acs.org/doi/abs/10.1021/ja047815i>; *J. Phys. Chem. C* 2018, 122, 22, 11961; <https://pubs.acs.org/doi/10.1021/acs.jpcc.7b12025>). We have added more solvent-dependent data (see updated Table S1 and Figure 3) to show how the PL energy shifts in different solvents with varying polarity index.

Revision: Updated Table S1 and Figure 3

Solvent	PL I	Polarity Index	Viscosity (cp r.t.)
DCM	670 nm	3.1	0.43
Toluene	650 nm	2.4	0.59
Hexane	610 nm	0.1	0.31
DCM	670 nm	3.1	0.43
1,2-Dichlorobenzene	660 nm	2.7	1.32
1-Butanol	600 nm	3.9	2.95
1-Octanol	590 nm	3.4	10.6

Figure 3. Solvent polarity and viscosity dependence, as well as the solid-state PL of Au₂₄. Comparison of the PL (A) and TCSPC (B) of Au₂₄ in DCM (black), toluene (blue) and hexane (red). TCSPC trajectories were detected from 550 to 650 nm. (C) Comparison of the PL of Au₂₄

in DCM (top), butanol (middle) and solid state (crystal or film, bottom). From top to bottom, the overall QY of Au₂₄ increases from 2 % to 30 %, which is accompanied by an increase in the ratio of PL I / PL II and the blue shift of the PL I. Inserted photographs: Au₂₄ in butanol and solid-state film and single crystals under UV lamp.

(5) These doubts on the origin don't affect the usefulness of the system for sensing, and the interesting optical data. Therefore I recommend major revisions.

Response: We thank for the reviewer's comments, and have provided major revisions here.

Reviewer #1 (Remarks to the Author):

I acknowledge the authors for answering thoroughly and pedagogically all my comments.

New Fig.2A is displayed now with the same color map scale ($\Delta T/T = -6$ to 4×10^{-4}) while in the previous version 2 color map scales were used to highlight the IR band (1050-1100 nm). However, the absolute value in $\Delta T/T$ for this IR band (1078 nm in Fig. 2B new version) is $\sim 2.5 \times 10^{-4}$ while it was $\sim 6 \times 10^{-4}$ (1078 nm in Fig. 2C previous version). The other two bands (601 and 790 nm) have similar trends. I am wondering thus if some additional treatments of data that justify these new color map scales (1030 nm to generate visible and near-IR probe light, therefore the scattering region between 900 and 1050 nm in the data map was removed, now I understand). Clearly, by looking at Fig. S1, I agree that there is no additional ESA "peak" at 1050 nm (while it was clearly present in the previous version...).

Thus I still don't understand why Fig.2A and B (in previous version) and Fig. 2A in the revision version are so different. And I still see some "dynamics" looking at $\Delta T/T$ at 601, 790 and 1078 nm wavelengths. Thus according to my opinion, the evolution of the $\Delta T/T$ curves can be still explained by the competitive relaxation processes between GSB \rightarrow PLI \rightarrow "PLIbis" and GSB \rightarrow PLII.

I agree that the apparent spectral structure (PLII) close to 1200 nm may result from the optical transmittance of DCM (~ 1150 nm).

The possibility of co-existing isomeric species has been excluded by using "high-purity" samples which are re-dissolved from their single crystals, and these single crystals are examined by single-crystal X-ray diffraction to confirm that no isomers exist inside. I agree that isomers usually show different absorption which can be easily distinguished by UV-Vis measurement, but in mixture, since absorption spectra are very broad, final features can be hard to distinguish. I am however convinced by their argument.

I am also convinced by their argument (additional TEM measurements) indicating that aggregation should not be the direct reason for the change of PL properties as long as the cluster structural distortion is hindered in solid states.

I also appreciate that the additional explanation and details on the high-pressure study (inducing an increase in the viscosity of the toluene solvent), which permits to rule out the transition from isolated Au₂₄ to "bulk-like" Au₂₄ under pressure.

Also, the new figure S15 permits to better evaluate the possibility of the solvent accessibility to Au kernel atoms.

In conclusion, except the excited-state dynamics of Au₂₄ presented in Fig.2 which is still not "unambiguous" (see my first comment), the authors have addressed solid arguments evidencing structural distortion and electron redistribution in gold nanoclusters. This paper can be accepted after minor revisions (see my first comment).

Reviewer #2 (Remarks to the Author):

This substantially revised version has satisfied my comments. I recommend publication.

Response to reviewers' comments

Reviewer #1 (Remarks to the Author):

I acknowledge the authors for answering thoroughly and pedagogically all my comments.

Response: Thanks for this comment!

New Fig.2A is displayed now with the same color map scale ($\Delta T/T = -6$ to 4×10^{-4}) while in the previous version 2 color map scales were used to highlight the IR band (1050-1100 nm). However, the absolute value in $\Delta T/T$ for this IR band (1078 nm in Fig. 2B new version) is $\sim 2.5 \times 10^{-4}$ while it was $\sim 6 \times 10^{-4}$ (1078 nm in Fig. 2C previous version). The other two bands (601 and 790 nm) have similar trends. I am wondering thus if some additional treatments of data that justify these new color map scales (1030 nm to generate visible and near-IR probe light, therefore the scattering region between 900 and 1050 nm in the data map was removed, now I understand). Clearly, by looking at Fig. S1, I agree that there is no additional ESA “peak” at 1050 nm (while it was clearly present in the previous version...). Thus I still don't understand why Fig.2A and B (in previous version) and Fig. 2A in the revision version are so different. And I still see some “dynamics” looking at $\Delta T/T$ at 601, 790 and 1078 nm wavelengths. Thus according to my opinion, the evolution of the $\Delta T/T$ curves can be still explained by the competitive relaxation processes between GSB \rightarrow PLI \rightarrow “PLIbis” and GSB \rightarrow PLII.

I agree that the apparent spectral structure (PLII) close to 1200 nm may result from the optical transmittance of DCM (~ 1150 nm). The possibility of co-existing isomeric species has been excluded by using “high-purity” samples which are re-dissolved from their single crystals, and these single crystals are examined by single-crystal X-ray diffraction to confirm that no isomers exist inside. I agree that isomers usually show different absorption which can be easily distinguished by UV-Vis measurement, but in mixture, since absorption spectra are very broad, final features can be hard to distinguish. I am however convinced by their argument. I am also convinced by their argument (additional TEM measurements) indicating that aggregation should not be the direct reason for the change of PL properties as long as the cluster structural distortion is hindered in solid states. I also appreciate that the additional explanation and details on the high-pressure study (inducing an increase in the viscosity of the toluene solvent), which permits to rule out the transition from isolated Au₂₄ to “bulk-like” Au₂₄ under pressure. Also, the new figure S15 permits to better evaluate the possibility of the solvent accessibility to Au kernel atoms.

In conclusion, except the excited-state dynamics of Au₂₄ presented in Fig.2 which is still not “unambiguous” (see my first comment), the authors have addressed solid arguments evidencing structural distortion and electron redistribution in gold nanoclusters. This paper can be accepted after minor revisions (see my first comment).

Response: Thank you very much for the comments. In the first version of the manuscript, the transient data for the pump at 1030 nm and the probe at 1050-1300 nm is presented as raw data (with intensity around -6×10^{-4} for probe at 1078 nm). However, for the probe at 500-900 nm and 1050-1300 nm, the experiment was setup differently, with different pump power, pump laser beam size (520 nm vs 1030 nm), which leads to a different pump fluence. The data shown in Fig 2A and Fig S1 in the new version of the manuscript is after the pump fluence has been corrected. Thanks to reviewer 1's comments in the first round of revisions, we realized that the first way we presented the data was misleading, so we corrected the data in second version of the manuscript. The correction consists of scaling the intensity, without changing the dynamics, based on the difference in pump fluence for the two pump lasers.

We agree with the reviewer that the excited state evolution is quite complex and rich. We are glad that the reviewer acknowledges that our experimental results show evidence for structural distortion and electron redistribution in gold nanoclusters, and the two-system model from PL I to PL II state is the most consistent with all the experimental results.

Revision: New details added in the experimental parts in page 10.

A 1030 nm laser was used to generate visible and near-IR probe light, therefore the scattering region between 900 and 1050 nm in the data map is removed. The experiment setup for the probes at 500-900 nm and 1050-1300 nm were different, and used different pump power and pump laser beam size. Therefore, corrections are made to the pump fluences in the reported data.